# Different Fiber Reinforcement Effects on Fly Ash-Based Geopolymer Long-Term Deflection in Three-Point Bending and Microstructure

**DOI:** 10.3390/ma15238512

**Published:** 2022-11-29

**Authors:** Rihards Gailitis, Leonids Pakrastins, Andina Sprince, Liga Radina, Gita Sakale, Krzysztof Miernik

**Affiliations:** 1Institute of Structural Engineering, Riga Technical University, Kipsalas 6A, LV-1048 Riga, Latvia; 2Department of Materials Engineering, Faculty of Material Engineering and Physics, Cracow University of Technology, Jana Pawła II 37, 31-864 Cracow, Poland

**Keywords:** fly ash-based geopolymer composite, long-term deflection, fiber-reinforced geopolymer

## Abstract

This study investigated the effect of a low amount of polyvinyl alcohol (PVA) and steel fiber reinforcement on fly ash-based geopolymer composite long-term deflection and its microstructure. For testing purposes, specimens with different amounts and types of fiber reinforcement as well as plain (reference) were prepared. The long-term deflection test was performed by loading specimens with 40% of the ultimate flexural strength. A microstructure analysis was performed using polished section specimens, and images were acquired at 25-times magnification on a scanning electron microscope. The results of the flexural strength test show that all geopolymer composites with fiber reinforcement have lower flexural strength than plain geopolymer composites. The long-term deflection tests show that the highest deflections exhibit 1% PVA fiber-reinforced specimens. The lowest amount of deflection is for 1% steel fiber-reinforced specimens. Specific creep shows similar results to plain, and 1% steel fiber-reinforced specimens, while 1% PVA and 0.5% PVA/0.5% steel fiber-reinforced specimen exhibits the same properties. The quantitative microanalysis of the polished section further confirms the deflection results. Specimens with 1% PVA fiber reinforcement have significantly higher porosity than all other specimens. They are followed by plain specimens and 1% steel fiber, and 0.5% PVA/0.5 steel fiber-reinforced specimens have almost the same porosity level.

## 1. Introduction

Currently, the most popular construction material is concrete based on ordinary Portland cement (OPC). Its popularity is mainly due to the low cost of concrete. As the worldwide population grows, it is estimated that the consumption of OPC will increase so much that yearly CO_2_ pollution will grow from around 7% at present to 17% [1].

Geopolymer is considered a very sustainable material, mainly because it can be produced from industrial waste materials. A significant number of studies show that geopolymer has the necessary properties to be a suitable construction material and is very likely to replace OPC concrete, completely in some cases [2,3,4,5,6]. There have been estimations that producing concrete-like materials by means of geopolymerization can reduce CO_2_ emissions regarding OPC production by up to 86% per one ton of Portland cement [1].

Geopolymer matrix can be produced from fly ash and various slags, such as granulated blast furnace slag, kaolin, and pozzolans. Most studies have researched geopolymers based on fly ash. Fly ash is a byproduct of coal power plants. In some countries, fly ash remains 20 to 60% cheaper than Portland cement. In most cases, these countries have coal power plants [7].

According to the life cycle assessment of ordinary Portland cement concrete (OPCC) and alkali-activated binary concrete (AABC), the AABC has 44.7% less kg CO_2_ eq/m^3^ than OPCC. These results indicate that AABC usage as an alternative to OPCC is valid [8].

As it is known, geopolymer composites have similar compressive strength to OPC-based composites. Geopolymer composite also achieves 85% of its strength in the first 48 h [9]. Moreover, its tensile and flexural strength are close to OPC but slightly more brittle. It is well known that fiber introduction into the composition of geopolymer composite in a certain amount reduces creep and shrinkage in compression and tension, further reducing cracking effects and redistributing stresses throughout the cross-section of the structure [10,11,12,13]. It is known that steel fibers have high mechanical strength, flexibility, and availability. They have many shapes and can be manufactured in different ways that further show their strength. The tensile strength of steel fibers differs from 310 to 2850 MPa [14]. The most popular polymer fibers are polyvinyl alcohol (PVA) and polypropylene (PP). Polypropylene fibers have low-cost favorable characteristics in high pH environments and the ability to control plastic shrinkage-caused cracking, but they have low thermal resistance, low modulus of elasticity, and poor interfacial contact with the cementitious matrix [15,16]. The PVA fibers have a higher modulus of elasticity and tensile strength, as well as showing higher chemical bonding with the cementitious matrix [17,18].

Tensile and flexural strength can be significantly increased by the addition of fibers. By adding 2% of sorghum fibers, the tensile strength can be increased by 36% [19]. It is claimed that the addition of 2% PVA fibers or 2% steel fibers, or hybrid fiber reinforcement consisting of 1% PVA and 1% steel fibers, leads to great flexural strength [20].

Long-term deflection assessment is of high importance for the further development of geopolymer construction structural design for serviceability. There are only a few deflection assessments under flexural stress results reported for geopolymer composites. Results from the research of [21] show a close correlation with OPCC’s long-term deflection properties. Still, data are inconclusive on whether geopolymer composite is subjectable to larger long-term deflections than OPCC.

The aim of this article is to determine the long-term deflection properties of different fiber-reinforced geopolymer composites under three-point bending and the fiber reinforcement influence on specimen microstructure after long-term and mechanical tests.

## 2. Materials and Methods

The geopolymer composite matrix was based on fly ash from the coal-powered power plant located in Skawina, Poland. This specific fly ash is suitable for geopolymer production because it contains spherical aluminosilicate particles. It is rich in oxides such as SiO_2_ (47.81%) and Al_2_O_3_ (22.80%). The significantly high content of SiO_2_ and Al_2_O_3_ in this fly ash is advantageous for geopolymerization.

Geopolymer composites preparations were made according to the following steps:1.Day 1—10 M NaOH solution preparation. Alkali solution is prepared by mixing NaOH flakes with water. As the chemical reaction is exothermic, the container with the solution after mixing is placed in cold water for one hour to reduce the temperature. Then, the R-145 sodium silicate is added to the sodium hydroxide and mixed until the solution has an even consistency. The solution is left until the next morning to settle.2.Day 2—Geopolymer specimen preparation.2.1.Specimen dry mix is prepared by mixing quartz sand and fly ash together. The sand and fly ash content ratios are 1:1 by mass. Sand and fly ash are mixed in a mixer for 5 min at the machine’s lowest speed.2.2.After dry mix preparation, the previously prepared alkali solution is added to the dry mix and mixed until the geopolymer achieves a moldable consistency. Mixing is performed for 15 min at the machine’s lowest speed [22,23]. For the fiber-reinforced specimens, after the first 15 min of mixing, the previously prepared and weighed fiber reinforcement (Figure 1) is added, and the whole geopolymer composition is mixed for 5 min. Whole geopolymer mixing is shown in Figure 2.2.3.The geopolymer mixture is poured into previously oiled plywood molds. The molds are vibrated to release entrapped air and covered with plastic film, then placed into a heat chamber at 75 °C for 24 h.

The weight and weight ratios regarding geopolymer composite preparation are compiled in Table 1. The properties of the fibers used in the specific geopolymer composites are presented in Table 2.

After mixing, the geopolymer composite was laid in plate molds and polymerized for 24 h at 75 °C. The polymerization process is shown in Figure 3.

After polymerization, plate shape specimens were cut into beam-shaped specimens with dimensions 20 × 75 × 450 mm. Specimen cutting and preparation were performed in Riga Technical University facilities (RTU). Specimens before and after cutting are shown in Figure 4.

After cutting, specimens were packed into plastic and aluminum film to prevent shrinkage. The aluminum plate was also glued to the specimen to allow accurate measurements of creep deflection. The procedure is shown in Figure 5.

After this, flexural strength was determined, and the specimens were placed on a deflection creep stand and loaded with 40% of the ultimate flexural strength value. Placement into the deflection creep stand was according to the scheme in Figure 6. The actual specimen placement into deflection creep stand is shown in Figure 7.

A creep test was carried out for 108 days in the RTU lab, and the specimens were unloaded on the 95th day of testing.

After the creep tests, three specimens were used for quantitative microstructure investigations and three to determine flexural strength after long-term testing, which were then also used for a microstructure analysis. The specimens intended for the microstructure analysis middle part where the load was applied and deflection measured were saturated with epoxy resin to develop polished section specimens for the microstructure testing purposes, as shown in Figure 8.

The polished section specimen development shown in Figure 8 and Figure 9 was performed according to the procedure mentioned in [24,25].

After the polishing process, the polished section specimens are examined using a scanning electron microscope JEOL IT200 (JEOL, Tokyo, Japan) to determine the loading effect on a measured zone of the beam specimens. Images of the specimens are taken at 25-times magnification.

## 3. Results and Discussion

The flexural strength was determined before and after the creep tests. For each mix type, three specimens were used each time to determine flexural strength values. The specimen age at the time of testing was 28 and 274 days. From the destructive deflection tests, the ultimate flexural load is determined, and the bending strength is calculated according to the equation:(1)$$\sigma =\frac{3FL}{2bd^{2}}$$
where:

*F*—Applied force;

*L*—Span of the specimen;

*b*—Width of the specimen

*d*—Thickness of the specimen

The flexural strength values are shown in Figure 10 and Table 3.

As shown in Figure 10 and Table 3, the plain geopolymer composites exhibit the highest flexural strength, followed by the 1% PVA fiber-reinforced specimens and 0.5% PVA/0.5% steel, and 1% steel fiber reinforced specimens with a 9%, 14%, and 23% reduction in flexural strength, respectively. Despite this factor, plain specimens have the highest standard deviation. For plain specimens, this is +/−0.86 MPa; +/−0.51 MPa for 1% PVA fiber-reinforced specimens; +/−0.20 MPa for 0.5% PVA/0.5% steel fiber-reinforced; and +/−0.14 MPa for 1% steel fiber-reinforced specimens. Similar flexural strength values were achieved by [26,27]. Nazari et al. studied boroaluminosilicate geopolymer with steel fiber reinforcement 2, 3, and 5% by volume, and the results show flexural strength ranging from 6.3 to 11.8 MPa. They found that this was achieved by not only increasing the fiber amount, but also by increasing the ratio of borax and sodium hydroxide ration. Still, the main influence on the flexural strength increase is the increase in fiber reinforcement. Plain geopolymer specimens exhibit flexural strength from 5.0 to 9.5 MPa [26]. Constancio Trindade et al. [27] tested geopolymer based on metakaolin reinforced with PVA and polyethylene (PE) fibers (the fiber content introduced in the geopolymers was 2% by volume). The flexural strength of the specimens that were not subjected to elevated temperatures showed significantly higher flexural strength than fly ash-based specimens. In other words, metakaolin geopolymer exhibits 19.7 MPa to PVA fiber-reinforced specimens and 23.5 MPa to PE fiber-reinforced specimens. Plain specimens have a flexural strength of 9.8 MPa. Others [28,29] report similar flexural strength with small-scale specimens that show a bending strength of 4.6 MPa to plain and 3.5 to 4.6 MPa reinforced with steel fibers. Large-scale beam-shaped specimens with reinforcement bars have bending strength from 22.46 to 29.36 MPa to reinforced specimens and 18.96 MPa to plain specimens. Still, according to the previously mentioned research and its specimen thickness-to-span ratio, it shows that the specimens tested in this study have rather remarkable flexural strength that most likely would be at its highest amount if the steel fiber reinforcement were approximately 4 to 5%, and the PVA fibers from 3 to 4%.

Specimens that were crashed after long-term tests showed a decrease in flexural strength. For specimens reinforced with 0.5% PVA/0.5% steel and 1% steel fibers, the decrease was slight, but for plain and 1% PVA fiber-reinforced specimens, the decrease was 14% and 15.2%, correspondingly.

For all of the geopolymer composite mixes, the flexural values determines at the age of 28 days are higher than [21] the 28-day flexural strength values. Plain GP flexural strength is 14.1% higher than the highest flexural strength achieved in [21]. Still, it has to be noted that the specimen dimensions here and in [21] are different.

Based on Table 3 and the claims of Z. Junwei [30] that appropriate fiber reinforcement should improve structural defects such as micro holes and micro cracks, this further leads to increased mechanical properties. In turn, this leads to the conclusion that the fiber amount used in this study is too low.

As seen in Figure 11, the lowest creep deflection is linked to the specimens with 1% steel fiber reinforcement, followed by plain geopolymer specimens, geopolymer with 0.5% PVA/0.5% steel fibers, and 1% PVA fiber-reinforced specimens. In other words, creep deflections for the plain specimens, 0.5% PVA/0.5% steel fiber-reinforced, and 1% PVA fiber-reinforced specimens are 48.2%, 53.1%, and 59.6% larger, respectively, than the 1% steel fiber-reinforced specimens.

If we compare the results in Figure 11 with the long-term deflection results from [31], we can see that, unlike OPC mortar specimens that would exhibit hydration effects and show that autogenous and drying shrinkage have a leading effect over creep effects, the GP specimens do not show such an effect. Furthermore, the long-term strains for the GP are 0.8% from the flexural strength registered to OPC mortar specimens tested at an early age. In their study, Un et al. [21] presented long-term deflection of composite geopolymer beams, and there is visible close relation with the Figure 11 curves. Furthermore, it is clear that when specimens reach the age of 100 days (in Figure 11, day 72), the creep strains stop generating and stabilize.

As flexural strength for different geopolymer compositions is different, the flexural deflections are also different. To evaluate creep deflections without different stress amount impact, the specific creep is calculated (see Figure 12). The calculation is made according to the equation:(2)$$\chi_{cr}\left(t,t_{0}\right)=\frac{\varepsilon_{cr}\left(t,t_{0}\right)}{\sigma }=\frac{\varepsilon_{kop}\left(t\right)-\varepsilon_{sh}\left(t\right)-\varepsilon_{el}\left(t,t_{0}\right)}{\sigma }=\frac{1}{E_{cr}\left(t,t_{0}\right)}$$
where:

$\chi_{cr}\left(t,t_{0}\right)$ is the specific creep,

$\varepsilon_{cr}\left(t,t_{0}\right)$ is the creep strain,

$\varepsilon_{kop}\left(t\right)$ is the total strain,

$\varepsilon_{sh}\left(t\right)$ is the shrinkage strain,

$\varepsilon_{el}\left(t,t_{0}\right)$ is the elastic strain,

σ is the compressive stress,

$E_{cr}\left(t,t_{0}\right)$ is the modulus of creep.

**Figure 12 materials-15-08512-f012:**
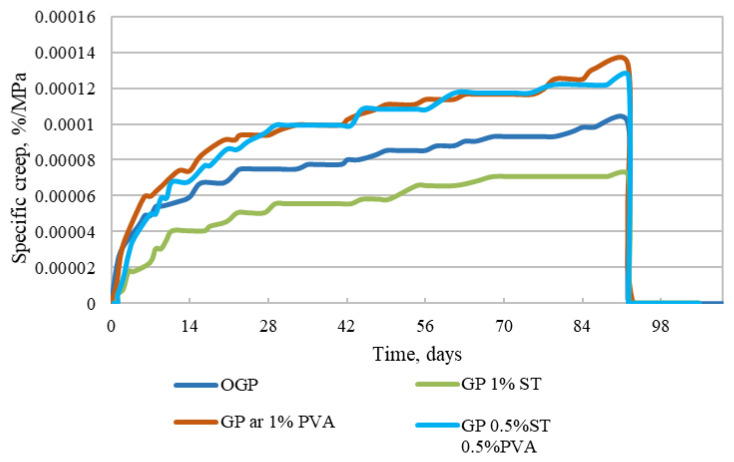
Plain, 1% PVA fiber-reinforced, 1% steel, and 0.5% PVA/0.5% steel fiber-reinforced specimen specific creep.

In Figure 12, the specific creep values show similar relations to the creep deflection values in Figure 11. The differences are shown in values. Specimens with 1% steel fiber reinforcement still have the lowest specific creep, followed by plain geopolymer specimens with 39.0% higher specific creep, 0.5% PVA/0.5% steel fiber-reinforced specimens with a 43.6% increase, and 1% PVA fiber-reinforced specimens with 52.2% higher specific creep. This leads to that the assumption that, even though specimens with the addition of polymer fiber—as mentioned in this study in Table 3 and other studies [26,27,28,29]—have higher flexural strength, they are more prone to creep effects and would have larger long-term deflections than steel fiber-reinforced specimens.

Further, to elaborate on the bending force influence on the specimen cross-section at which the load is applied, a SEM image quantitative analysis is carried out. In Figure 13, an acquired SEM image dividing the sequence into layers is shown on the specimen that is subjected to the flexural strength test.

The quantitative image analysis is based on a determination of the specimen cross-section composition-parameter quantity ratio to all areas of the cross-sections. In other words, each partition of the studied cross-section is divided into a separate layer, and a specific RGB color is assigned to this layer. When all of the images of the specific cross-section are divided into layers and color codes are assigned, the number of pixels is counted for all of the raw undivided image, as well as for each of the specific color layers. Then, the acquired number of pixels linked to the specific layer is attributed to the total number of pixels in the whole image, and the quantitative amount of the specific cross-section composition partition is determined. In the case of the long-term deflection specimens, the zone viewed in this research (the location in which the load is applied) is only 75 mm thick. As each of the polished section specimens at the beginning is approximately 15 mm thick, only four specimens are made and studied. The results of the quantitative image cross-section analysis for the specimens used for long-term deflection tests are compiled in Table 4, and in Table 5, the results from the analysis of flexural strength regarding the specimen surface quantitative image are compiled.

First, from Table 4, it is clear that the highest quantity of air voids was in the plain geopolymer composite specimens. From the fiber-reinforced specimens, it becomes apparent that the highest air void amount to the amount of fibers visible in the cross-section was from the 1% steel fiber-reinforced specimens. The smallest ratio between fiber and air void amount was linked to the specimens reinforced with 1% PVA fibers.

Furthermore, as all of the reinforced geopolymer composites were 1% reinforced from the filler and matrix mass, it is clear that the PVA fiber reinforcement amount was around three times larger than the amount of steel fibers that were used, and about two times larger than the PVA and steel fiber reinforcement mix. The previous two statements lead to the assumption that steel and PVA + steel fiber reinforcement does not have such a significant frothing capability while mixing as the PVA fibers alone. The reduction in the matrix for the PVA fiber specimens is significant.

In the microstructure images, cracks can be observed from the specimens that were destroyed during the flexural strength tests. In these cases, the fiber-bridging effect is noticeable, especially for the specimens with 1% PVA fiber incorporation. Furthermore, the crack opening for specimens reinforced with 1% PVA fibers is bigger than for all the other specimens. This correlates with the study of Y. Hiddaji et al. [32], in which SEM was used to determine the microstructure changes prior to and after high-temperature exposure in metakaolin and phosphate sludge-based geopolymer composites reinforced with glass fibers. In specimens that were not exposed to a temperature impact, they found small microcracks caused by water evaporation. They also found larger crack distribution of the glass fiber-reinforced specimens than the plain specimens that had straight cracks on the fractured surface. This shows that fiber reinforcement, due to polymerization effects and water evaporation, creates small but quantitatively more cracks than in plain specimens. Furthermore, Z. Deng et al. [33] found that PVA–fiber interaction with geopolymer composites leads to an increase in porosity. Furthermore, the authors indicated that with a higher PVA fiber content, higher porosity would be achieved. They found that when the PVA fiber amount is increased to 0.3% and up to 0.6% increases by 7.01% and 9.13% from the plain specimens. Thus, we can assume that the plate specimens used in this study had much better entrapped air release, unlike the prismatic specimens used by Z. Deng.

From Table 5, it becomes apparent that the changes in the polished section surface composition in contrast to Table 4 are significant. The specimens reinforced with 1% PVA fibers have the highest crack and air void amount and are followed by plain specimens, 1% steel fiber-reinforced specimens, and 0.5% PVA/0.5% steel fiber-reinforced specimens. Therefore, it is safe to assume that even though 1% PVA fiber-reinforced specimens have the lowest flexural strength, they retain some load-bearing capacity for the longest period, while the load is applied, and cracking of the stretched zone has begun. This is less so for 1% steel fiber-reinforced specimens, and less again for 0.5% PVA/0.5% steel fiber-reinforced specimens. This was also observed while specimens were loaded. The period between the achievement of ultimate flexural strength value and specimen collapse was longer than the 1% PVA fiber-reinforced specimens, while for 1% steel fiber-reinforced and 0.5% PVA/0.5% steel fiber-reinforced specimens, it was similar. Furthermore, it can be noted that, as expected, the plain specimens that collapsed under load were brittle, while the fiber-reinforced specimen collapse was plastic. In this case, PVA fibers that have a lower modulus of elasticity than steel fibers show that they cannot provide stress distribution throughout the specimens. Moreover, this leads to the assumption that the material is more elastic, and it would deflect other specimens with different or no reinforcement until it reaches the ultimate bending strength. This leads to bigger cracks and empty areas in the specimen cross-section, as is visible in Table 5 and when comparing with Table 4.

## 4. Conclusions

The long-term deflection properties of fly ash-based geopolymer composites reinforced with PVA and steel fibers were determined in a 109-day long creep test. Flexural strength was determined before and after the deflection creep tests, and a quantitative microstructure analysis was conducted on the specimens that were exposed to both the long-term test and a flexural strength test after the long-term test. The results of the performed tests show that:The highest flexural strength before and after the tests is found in the plain geopolymer specimens. The 1% PVA fiber-reinforced and 0.5% PVA/0.5% steel fiber-reinforced specimens have 9% and 14.2% lower flexural strength, respectively. The 1% steel fiber-reinforced specimens have 23.2% lower flexural strength than the plain specimens. After long-term deflection tests, the flexural strength values from the plain specimens are 11.1% and 11.4% lower for 1% PVA and 1% steel fiber reinforcement and 0.9% higher for the specimens reinforced with 0.5% PVA/0.5% steel fibers;The lowest long-term deflection in three-point bending is linked to the 1% steel fiber reinforced specimens. The 0.5% PVA/0.5% steel fiber-reinforced, plain, and 1% PVA fiber-reinforced specimens have 49.3%, 51.7%, and 60.6% higher long-term deflection, respectively;Specimens reinforced with 1% steel fibers or those with the lowest specific creep are less likely to deform under three-point bending, followed by plain specimens, 0.5% PVA/0.5% steel fiber-reinforced specimens, and 1% PVA fiber-reinforced specimens, by 37.1%, 43.3%, and 53.5%, respectively;The microstructure quantitative analysis of long-term deflection test specimens shows that specimens with 1% PVA fiber reinforcement have two times higher detected fiber reinforcement amount than all the other fiber-reinforced specimens, while air void amounts for the 1% PVA fiber reinforced specimens are 1.51 and 1.37 times higher than 0.5% PVA/0.5% steel, and 1% steel fiber-reinforced specimens. The flexural strength test specimen microstructure quantitative analysis showed 5.52% and 7.50% lower air void and crack amount linked to 1% steel fiber and 0.5% PVA/0.5% steel fiber-reinforced specimens than to 1% PVA fiber reinforced specimens, showing that these specimens have lower plasticity than those reinforced with 1% PVA fibers;Overall, it is apparent that beam-shaped specimens with random fiber distribution throughout have lower deflection strength. Moreover, steel fiber incorporation into geopolymer specimens seems to be most beneficial for reducing long-term deflection. It can be concluded that for the bent specimens, not only the lowest bending strength and its reduction while exposed to the long-term load application, but also the lowest long-term deflection and specific creep is linked to the specimens with 1% steel fiber incorporation.

The next stage of this research will be to take all the data from the composition geopolymer composite long-term tests on compression, tension, and three-point bending and develop a model on long-term property assessment for this kind of geopolymer composite, as well as to identify the links and similarities between the tensile and compressive long-term properties and long-term flexural properties of these geopolymer composites.

## Figures and Tables

**Figure 1 materials-15-08512-f001:**
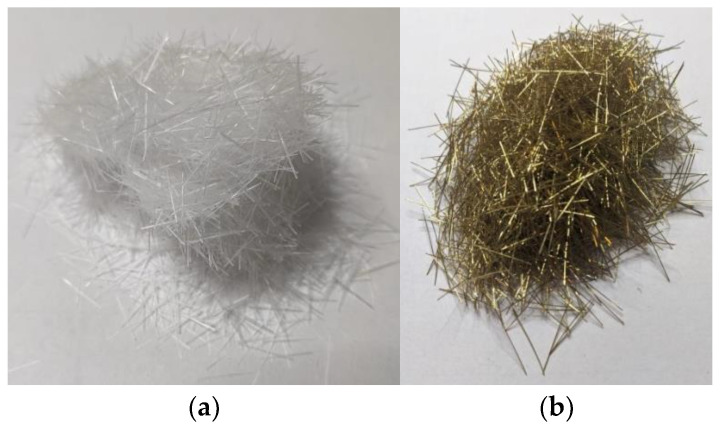
Used PVA (**a**) and steel (**b**) fibers.

**Figure 2 materials-15-08512-f002:**
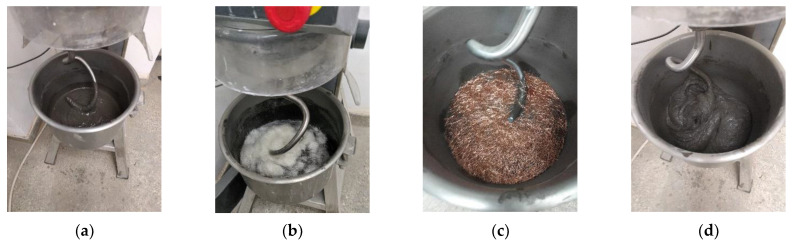
Geopolymer composite composition preparation procedure from geopolymer paste (**a**) to PVA (**b**) and steel (**c**) fiber addition and end composition (**d**).

**Figure 3 materials-15-08512-f003:**
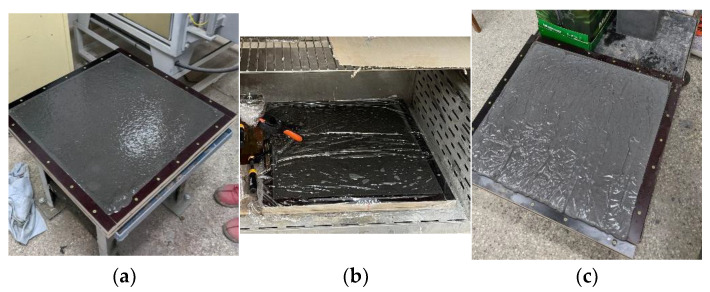
Geopolymer composite molding (**a**) and polymerization process (**b**,**c**).

**Figure 4 materials-15-08512-f004:**
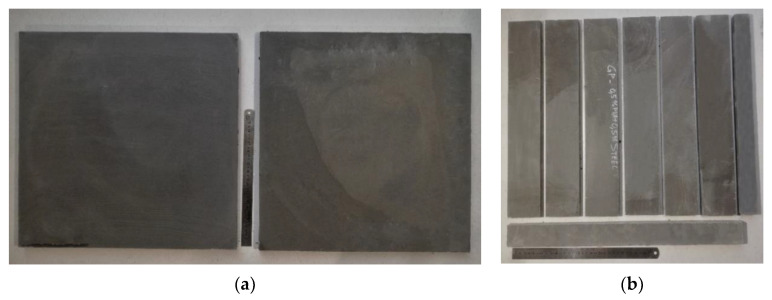
Geopolymer composite plate specimens before (**a**) and after (**b**) cutting.

**Figure 5 materials-15-08512-f005:**
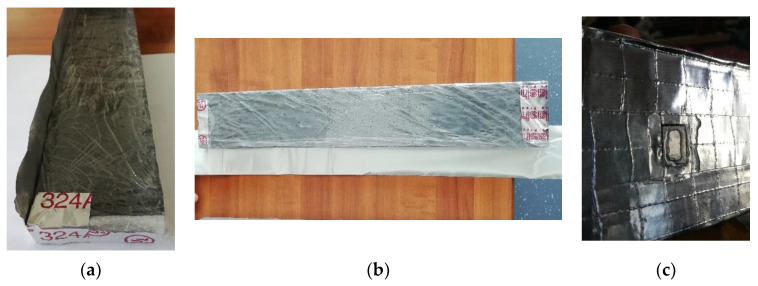
Geopolymer composite specimen wrapping (**a**,**b**) and plate-gluing (**c**) process.

**Figure 6 materials-15-08512-f006:**
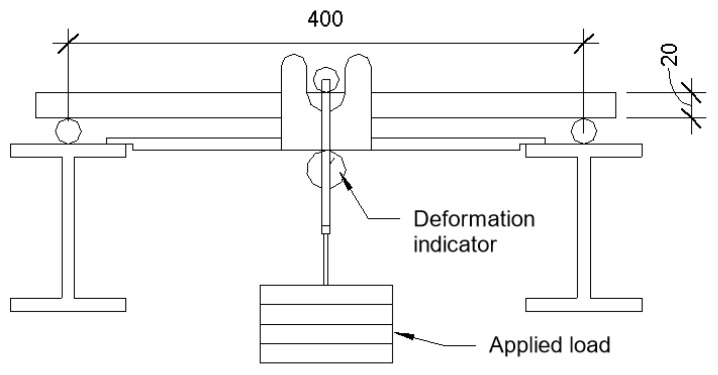
Specimen placement into the long-term deflection test stand.

**Figure 7 materials-15-08512-f007:**
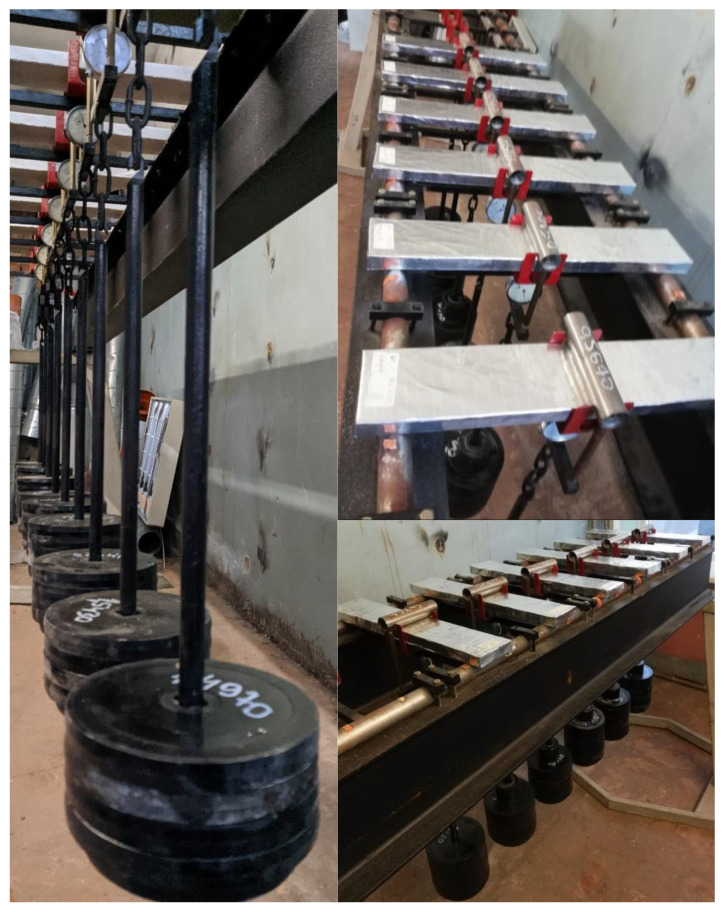
Setup and testing of long-term deflection strain in three-point bending.

**Figure 8 materials-15-08512-f008:**
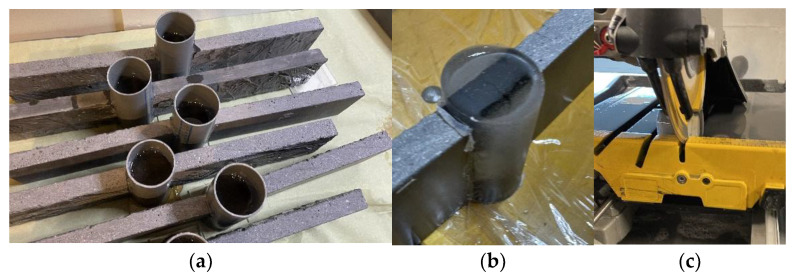
Geopolymer composite polished section casting into epoxy (**a**,**b**) and cutting (**c**) procedure.

**Figure 9 materials-15-08512-f009:**
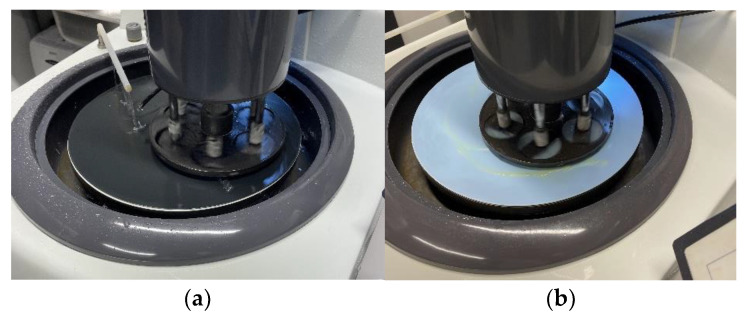
Geopolymer composite polished section specimen polishing with sandpaper (**a**) and polishing paste (**b**).

**Figure 10 materials-15-08512-f010:**
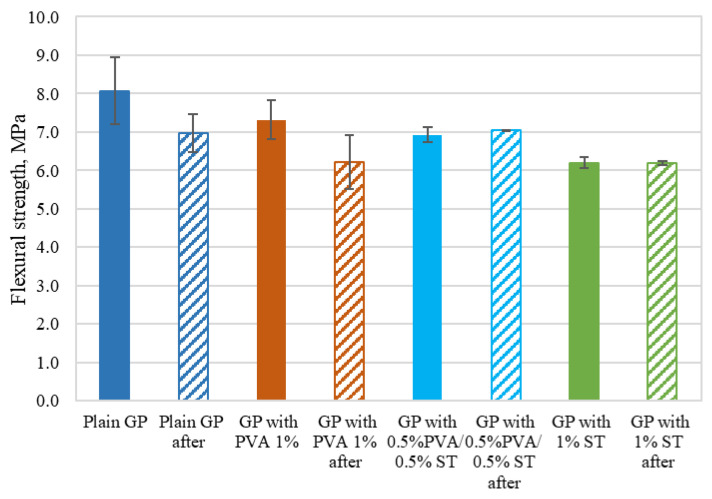
Geopolymer composite flexural strength values with measurement errors before and after long-term deflection tests.

**Figure 11 materials-15-08512-f011:**
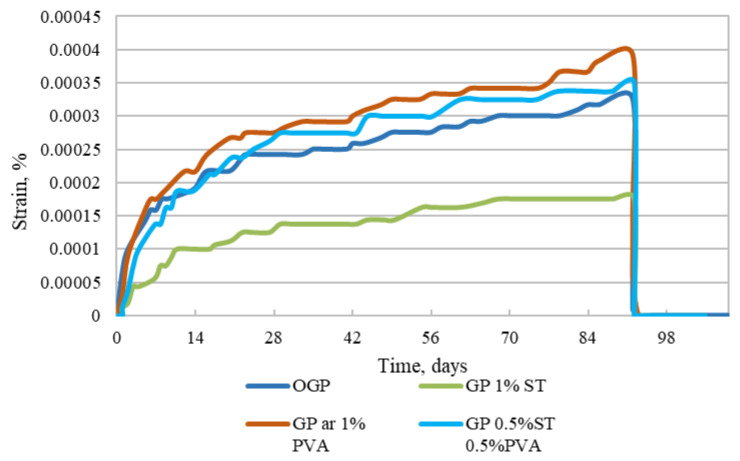
Plain, 1% PVA fiber-reinforced, 1% steel and 0.5% PVA/0.5% steel fiber-reinforced specimen creep deflections.

**Figure 13 materials-15-08512-f013:**
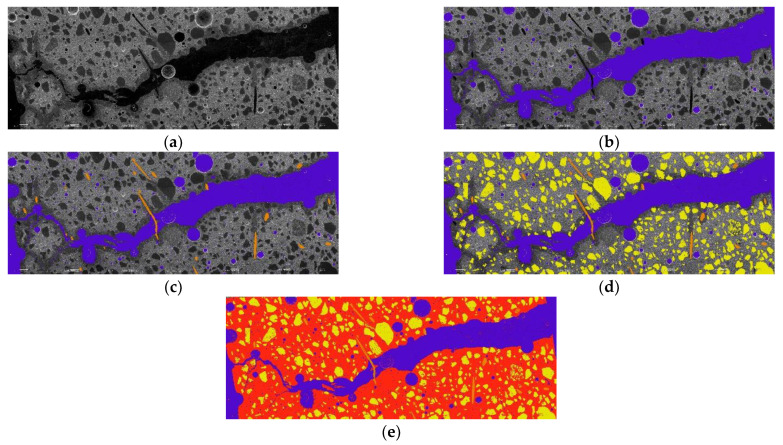
Geopolymer specimen reinforced with 0.5% PVA/0.5% steel fibers after long-term and bending strength test image dividing sequence from raw image (**a**) to air voids (**b**), added reinforcement (**c**), and filler (**d**) and matrix (**e**) in 25-times magnification. The image area is 20 mm high and 75 mm wide.

**Table 1 materials-15-08512-t001:** Used geopolymer composite alkali solution and dry mix quantitative parameters.

Alkali Solution		Dry Mix	
Constituent	Weight (g)	Constituent	Weight Ratio
NaOH flakes	400	Quartz sand	1.00
Water	1000	Fly ash	1.00
R-145 Na_2_O + SiO_2_ solution (molar module 2.5, density 1.45 g/cm^3^)	3500	Fibers	0.01

**Table 2 materials-15-08512-t002:** Basic properties of the used fibers.

Fiber Parameter	PVA Mesofibers (MasterFiber 400/401)	Steel Fibers (La Graminga GOLD)
Length (mm)	18.00	20.00
Diameter (mm)	0.16	0.30
Tensile strength (MPa)	790–1160	2635–3565

**Table 3 materials-15-08512-t003:** Geopolymer composite flexural strength average values and coefficient of variation values.

Geopolymer Composite Type	Age 28 Days		Age 274 Days	
	Flexural Strength (MPa)	Coefficient of Variation	Flexural Strength (MPa)	Coefficient of Variation
Plain GP	8.07	10.67	6.98	7.05
1% PVA GP	7.32	6.93	6.21	11.25
0.5% PVA/0.5% St GP	6.93	2.85	7.05	1.34
1% Steel GP	6.20	2.27	6.18	0.87

**Table 4 materials-15-08512-t004:** Results of geopolymer composite polished-section microstructure image quantitative analysis after long-term deflection test.

Geopolymer Composite Type	Matrix (%)	Filler (%)	Air Voids (%)	Fiber Reinforcement (%)
Plain GP	75.93	19.05	5.02	-
1% PVA GP	74.58	19.62	4.64	1.16
0.5% PVA/0.5% St GP	77.76	18.12	3.65	0.47
1% Steel GP	77.50	17.28	4.62	0.60

**Table 5 materials-15-08512-t005:** Geopolymer composite from flexural strength test polished-section microstructure image quantitative analysis results.

Geopolymer Composite Type	Matrix (%)	Filler (%)	Air Voids, Cracks (%)	Fiber Reinforcement (%)
Plain GP	65.10	16.33	18.57	-
1% PVA GP	61.02	16.04	22.00	0.94
0.5% PVA/0.5% St GP	69.09	15.99	14.50	0.42
1% Steel GP	67.87	15.12	16.48	0.52

## Data Availability

The authors consider that the data presented in this study are as they are. At the request raw data can be provided. Also compressive strength test results can be seen in this publication of the tested geopolymer mix https://doi.org/10.3390/cryst11070760.

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
