# Peer review of "Different Fiber Reinforcement Effects on Fly Ash-Based Geopolymer Long-Term Deflection in Three-Point Bending and Microstructure"

_materials, 2022, doi:10.3390/ma15238512_

Round 1

Reviewer 1 Report

This topic studies the influence of different fiber reinforcement on fly ash-based geopolymer composite and a lot of work has been done. While there are still many problems. In particular, many contents in the article are not stated clearly, such as test methods, data analysis methods, etc. The authors are suggested to further revise the manuscript. Some suggestions:

1. The Abstract is needed to be reorganized and some elements are not well displayed. For example, the introduction of the background needed to be supplemented. Try to make it more concise. And some information is not correct, such as In total four geopolymer mixes were prepared

2. The introduction section needs to be further enriched. There are few introductions about fiber reinforcement.

3. Line 52, “results show a close correlation”, who does have a close correlation to the long-term deflection properties? These statements should be more specific.

4. Table 1 is not well displayed.

5. Try to improve the presentation of the Figure 10 and Figure 11. Actually, Figures a and b in Figures 10 and 11 can be displayed together.

6. Why are the pictures in Figure 12 not arranged horizontally?

7. The article lacks specific descriptions of many tests. For example, line 102,some are used to determine flexural strength after long-term testing”, but how to do this long-term testing? The experimental instruments and methods should be further detailed. In the same case, how to get flexural strength, specific creep shown in Figure 11, etc.? Please provide calculation formula or determination methods.

8. How to determine the data results in Table 2 and 3?? The analysis process based on SEM images needs to be supplemented.

Author Response

Thank you for your review. Please see the answers in the uploaded word document and see the revised manuscript.

Reviewer 2 Report

The research topic is Different Fiber Reinforcement Effect on Fly Ash Based Geopolymer Long-Term Deflection in Three-Point Bending and Microstructure.

The research topic is highly topical and combines interesting research areas. 

Specifically, it is Geopolymer composite and fiber composite, where electron microscopy is also used.

The basic structure of the manuscript is well chosen. However, a discussion chapter should be included. 

However, before the contribution is published, it is necessary to improve the overall presentation and quality of the results from the presented research.

Recommendations and comments on the manuscript:

1) Improve the Introduction section. A number of authors are devoted to the research area, when it is necessary to focus on the originality and novelty of the topic. It is also necessary to provide enough information about the current state. It is important to be interested in the durability and service life of the composite for the solved area. The mentioned area also includes resistance to aggressive substances, ant tolerance, etc. There is a whole range of research in the addressed area, for example https://doi.org/10.3390/su13020473 ; https://doi.org/10.3390/ma14154264

2) Provide a link (reference) to the fibers used. Were the properties tested? or are they taken from the manufacturer's data sheet.

3) State in more detail the composite production procedure in the points in the text.

4) Figure 6. Improve the description and text in the image - processing quality

5) State the geopolymer composite recipe more clearly in the table

6) Add list the results for 28 and 274 days separately .

7) Give the coefficient of variation for the results.

8) Provide a picture with a bending test diagram.

9)Figure 12 - Specify the size of the image area (mm x mm) in the label

10) Units are given in brackets (m3)

11) It is necessary to include a discussion chapter where the results will be discussed in detail.

12) Improve the conclusion chapter

The authors tackle an interesting topic. The experimental program is also interesting. However, the manuscript must be improved before publication.

Author Response

(The authors gave the same response as above.)

Reviewer 3 Report

Paper ID: materials-2042787

Type: Article 
Title: 
Different Fiber Reinforcement Effect on Fly Ash Based Geopolymer Long-Term Deflection in Three-Point Bending and Microstructure

Authors: Rihards Gailitis , Leonids Pakrastins , Andina Sprince , Liga Radina , Gita Sakale , Krzysztof Miernik

This study investigates the effect of a low amount of polyvinyl alcohol (PVA) and steel fiber reinforcement on fly ash-based geopolymer composite long-term deflection and its microstructure. Although the testing methods and compared results attained in the present study show the importance of the paper, The authors should address the following comments: 

  1. Novelty in comparison to recent literature? Need to be emphasized.
  2. The results in the paper might be more discussed by the relevant literature.
  3. Introduction section: You should focus on geopolymer. A clear difference between geopolymer and alkali-activated materials has been established in the literature. The authors should discuss these materials further by considering this distinction.
  4. Line 33: in my opinion 17% CO2 emission for OPC is too much. Please justify this with other literatures.
  5. Introduction section is not enough for a scientific paper. I strongly suggest that the authors should be rewritten this section.
  6. Materials and Methods: The authors should add the chemical composition of aluminosilicate precursors.
  7. Line 65: The authors chose 10-molar (10M) NaOH and combined with the 2.5 M sodium silicate. Why?
  8. Please replace “x” with “×” throughout the manuscript.
  9. Methods section is missing. The authors should give the detailed methods.
  10. There should be a space between number and unit. Please correct these errors in the paper.
  11. Throughout the text, there are some typos that must be eliminated.
  12. The conclusion part seems to be more like an experimental report rather than a scientific paper. I strongly suggest for authors present their conclusions more concisely, avoiding repetition of the obvious and simple results.

Author Response

(The authors gave the same response as above.)

Round 2

Reviewer 1 Report

The manuscript has been well revised. It is recommended to be accepted for publication.

Author Response

Dear reviewer, 

Thank you for your review.

Reviewer 2 Report

The authors improved the manuscript. 

However, the quality and discussion of results from the experimental program still needs to be improved. 

It is necessary to fundamentally improve the discussion of the results. There are many researches and publications in the addressed area of research and it is necessary to state what is new about the presented research and results.

Author Response

Dear reviewer,

Thank you for your review. Additional discussion is added to the manuscript.

Reviewer 3 Report

The authors have made the necessary changes Therefore the manuscript can be accepted after the following minor concerns:
1. Table 1: Please remove the minor compounds.

2. Why did the authors use the chemical composition of the study [22]? It is very easy to determine oxides.

3. If the authors are unable to perform chemical oxide analysis, I recommend removing Table 1. Because the fly ash used in this study has  SiO2 (47.81%) and Al2O3 (22.80%).

Author Response

Dear reviewer,

Thank you for your review. According to your comments, Table 1 is removed from the manuscript.
